# PET/CT targeted tissue sampling reveals virus specific dIgA can alter the distribution and localization of HIV after rectal exposure

Roslyn A. Taylor[1], Sixia Xiao[1], Ann M. Carias[1], Michael D. McRaven[1], Divya N. Thakkar[1], Mariluz Araínga[2], Edward J. Allen[1], Kenneth A. Rogers[2,3], Sidath C. Kumarapperuma[4], Siqi Gong[2,5,6], Angela J. Fought[7¤a], Meegan R. Anderson[1], Yanique Thomas[1], Jeffrey R. Schneider[1¤b], Beth Goins[4†], Peter Fox[4], Francois J. Villinger[2,3], Ruth M. Ruprecht[2,3,5,6], Thomas J. Hope[1]*

1 Department of Cell and Developmental Biology, Northwestern University Feinberg School of Medicine, Chicago, Illinois, United States of America, 2 New Iberia Research Center, University of Louisiana at Lafayette, Lafayette, Louisiana, United States of America, 3 Department of Biology, University of Louisiana at Lafayette, Lafayette, Louisiana, United States of America, 4 Research Imaging Institute, University of Texas Health San Antonio, San Antonio, Texas, United States of America, 5 Department of Microbiology, Immunology, and Molecular Genetics, University of Texas Health San Antonio, San Antonio, Texas, United States of America, 6 Texas Biomedical Research Institute and Southwest National Primate Research Center, San Antonio, Texas, United States of America, 7 Department of Preventative Medicine, Division of Biostatistics, Northwestern University Feinberg School of Medicine, Chicago, Illinois, United States of America

† Deceased.
¤a Current address: University of Colorado Denver, Denver, Colorado, United States of America
¤b Current address: Rush University, Chicago, Illinois, United States of America
* thope@northwestern.edu

**Data Availability Statement:** All relevant data are within the manuscript and its Supporting Information files.

## Abstract

Human immunodeficiency virus (HIV) vaccines have not been successful in clinical trials. Dimeric IgA (dIgA) in the form of secretory IgA is the most abundant antibody class in mucosal tissues, making dIgA a prime candidate for potential HIV vaccines. We coupled Positron Emission Tomography (PET) imaging and fluorescent microscopy of $^{64}$Cu-labeled, photoactivatable-GFP HIV (PA-GFP-BaL) and fluorescently labeled dIgA to determine how dIgA antibodies influence virus interaction with mucosal barriers and viral penetration in colorectal tissue. Our results show that HIV virions rapidly disseminate throughout the colon two hours after exposure. The presence of dIgA resulted in an increase in virions and penetration depth in the transverse colon. Moreover, virions were found in the mesenteric lymph nodes two hours after viral exposure, and the presence of dIgA led to an increase in virions in mesenteric lymph nodes. Taken together, these technologies enable *in vivo* and *in situ* visualization of antibody-virus interactions and detailed investigations of early events in HIV infection.

## Author summary

Vaccines provide protection in humans by eliciting the production of antibodies when exposed to a specific pathogen. Currently, an effective human immunodeficiency virus

**Funding:** This work was supported by the National Institutes of Health grants: NIH P01 AI48240 (TJH, SCK, BG, PF, FV, RMR) and NIH P50 AI150464 (TJH), 1 K01 OD026571-01 (AMC), and K01OD024882-01 (JRS). The funders had no role in study design, data collection and analysis, decision to publish, or preparation of the manuscript.

**Competing interests:** The authors have declared that no competing interests exist. Author Beth Goins was unable to confirm their authorship contributions. On their behalf, the corresponding author has reported their contributions to the best of their knowledge.

(HIV) vaccine does not exist. Since the beginning of the HIV epidemic, approximately 38 million people have died, creating the need to develop an HIV vaccine. The most common antibody in the organs that are exposed to HIV is dimeric IgA (dIgA). Here, we used multiple imaging techniques to determine how HIV travels throughout the colon once introduced into the body and how dIgA influences HIV movement in the rectum. We found that dIgA increased the amount of HIV found in the colon, the distance it travelled, and the depth into tissues that HIV penetrated. dIgA also increased the amount of HIV in the mesenteric lymph nodes two hours after viral exposure. Our study shows these imaging technologies can be used to examine interactions between viruses and antibodies in early HIV infection in the natural context of the anatomy and physiology of the rhesus macaque model.

## Introduction

Vaccines can protect hosts from infection by directing the production of antibodies that form immune complexes with antigens[1]. While there is currently no effective vaccine for the human immunodeficiency virus (HIV), one that elicits a response in the IgA class might be a strong candidate given the high concentration of IgA in colorectal tissue, a primary site for HIV transmission. IgA is found primarily in the mucosa as a dimer (dIgA). In humans, IgA has two subclasses, IgA1 and IgA2, which differ structurally from each other in their hinge region[2,3]. In the gut, IgA1 is primarily found in the small intestine, while IgA2 is found in both the small intestine and the colon[4,5]. Both IgA1 and IgA2 have the ability to bind and sequester pathogens, creating the need to study the role of both in potential protection from HIV rectal transmission.

IgA plays an important immunological role[6,7]; its primary function in the mucosa is to prevent further infection by immune exclusion[6]. In this process, dIgA binds pathogens while associating with mucus, cell associated mucins, and other factors in the glycocalyx, thus trapping and excluding the pathogen's ability to enter underlying tissues. Through this mechanism, the virus is effectively sequestered within immune complexes. This physical exclusion can extend to intracellular vesicular trafficways where particles entering from the lumen by transcytosis through columnar epithelial cells can interact with IgA resulting in immune complexes that are shuttled to either the basal or apical surface of the epithelial barrier[8]. Trafficking to the apical surface would lead to shedding of the virus via mucus clearance. For example, studies have illustrated that IgA is able to interact with HIV within epithelial cells, forming a virus-antibody complex that can be transported to the lumen, thus preventing replication[9–11]. These IgA-pathogen immune complexes can also be transported through the mucosal barrier to be recognized by antigen-presenting cells that mediate their clearance and presentation to relevant T cells[12]. These antigen-presenting cells also transport these immune complexes to draining lymph nodes to facilitate interaction with various immune cells to mediate and mature immune responses.

Notably, most vaccines do not stimulate protective IgA responses. To date, robust IgA responses have been elicited by oral vaccines, such as the attenuated poliovirus and rotavirus vaccines[13,14]. For example, in the RV144 HIV vaccine clinical trial, a serum monomeric IgA anti-HIV response was associated with an increased risk of HIV acquisition in correlates analysis[15]. This observation raised questions about the underlying mechanisms supporting this correlate of an inefficient or aberrant response to the RV144 vaccine associated with the absence of protection[15]. On the other hand, evaluation of serum IgA in the more recent

HVTN 505 clinical trial demonstrated that there was no correlation between serum Env IgA levels and HIV acquisition[16]. Clearly, a better understanding of how vaccination can stimulate IgA responses and the impact of these responses on HIV acquisition require additional analysis.

To gain insights into the ability of dIgA antibodies to mediate protection from mucosal challenge, we previously developed a non-human primate (NHP) model to examine the potential role mucosal IgA has in protection against rectal challenge with SHIV-1157-ipEL-p[17], a clade C, tier 1 CCR5-tropic virus. Through investigating the role of IgG1, dIgA1, and dIgA2 isotypes of the HIV-specific HGN194 monoclonal antibody specificity[18] in SHIV-1157-ipEL-p rectal acquisition in rhesus macaques (RMs), we found that HGN194 dIgA1, rectally applied 30 minutes before rectal challenge provided 83% protection (5 out of 6 animals), whereas HGN194 dIgA2 only protected 17% of the RMs (1/6)[19]. In a later study, we found that intravenous HGN194 IgG1 given at a low dose provided 0% protection (0/6). Unexpectedly, giving both intravenous low-dose HGN194 IgG1 and rectally applied HGN194 dIgA2 treatments afforded 100% protection (6/6)[20]. Despite the promise provided by these observations of IgA function to prevent rectal transmission in a RM model, mechanistic insights to the role mucosal dIgA could play in mediating the prevention of HIV acquisition when combined with an IgG that cannot block infection, but instead causes a slight delay in the peak viral load. Understanding these mechanisms has the potential to lead to improved protective therapeutics and vaccines.

To gain better insights into how dIgA affects HIV acquisition *in vivo*, we sought to develop a RM model that would allow us to monitor the distribution and localization of both dIgAs and viral particles under the conditions of the HGN194-dIgA/IgG1 studies described above and shown to provide protection. To provide a highly sensitive method to monitor the localization of virus and antibodies we decided to evaluate an approach utilizing "beacon targeted necropsy." This type of necropsy is largely inspired by our recent success in the identification of early foci in mucosal SIV acquisition utilizing luciferase expression to guide necropsy to pinpoint small pieces of tissues with robust luciferase signal[21]. This luciferase beacon is generated from a dual reporter SIV-based vector, which is mixed with the challenge inoculum, and allows for the identification of sites where particles can overcome innate mucosal barriers to reach the appropriate target cells, thereby facilitating the study of SIV infection within tissues at the portal of transmission. However, there is a major caveat with this methodology, which is these photon-based beacons are limited by opaque tissues.

In contrast, radioactivity-based imaging approaches overcome the limitations of photons in NHP studies because the detection of the radioactive signal is minimally impacted by tissue opaqueness. Therefore, positron emission tomography (PET) coupled with computerized tomography (CT) allows an unbiased visualization of the radiolabeled molecule at the whole animal level with the additional ability to have multiple scans to provide insights into kinetics and dynamics of distribution. Interpretation of the PET signal can be facilitated by the coupling of an imaging modality that reveals anatomical structures, such as bones and tissues, and utilizing imaging approaches, such as 3D x-rays, through computerized tomography (CT) or the use of magnetic resonance imaging (MRI). Again, we can utilize this PET detected beacon, now as a signal of radioactivity, as a guide to identify small tissue pieces which contain the labeled molecules of interest. The signal is then validated by fluorescence microscopy methods. Here, we fluorescently label the dIgA by direct conjugation with Cy5 and utilize our previously developed system of HIV containing a photoactivatable (PA) form of green fluorescent protein (GFP) which allows the visualization of individual intact viral particles in tissue samples as previously described [22–26]. The ability to efficiently load HIV-BaL with a PA-GFP-VPR and the ability of the HGN194 binding site specificity to neutralize HIV-BaL make this well-

defined virus system an acceptable substitute for the SHIV-1157ipEL-p utilized in the challenge studies [20]. The inability of the HIV-BaL to infect macaque cells is not relevant in the short, two-hour duration of virus distribution analyzed. In the studies presented here, we utilized both of these technologies and the RM rectal challenge model to determine whether the rectally applied dIgAs can influence HIV interactions with rectal and colonic mucosal barriers. First, we intrarectally introduced either dIgA1 or dIgA2 and a fluorescently tagged virus that was radioactively labeled with [64]Cu. We then studied the tissue localization of dIgA and virus in an unbiased manner using state-of-the-art non-invasive imaging technologies. To do this, we used whole-body PET coupled with either magnetic resonance imaging (MRI) or CT, followed by fluorescent microscopy of a photoactivatable-GFP HIV-BaL (PA-GFP-BaL)[23,27–29]. Utilizing both of these novel techniques at two hours post-challenge, we find that the administered dIgA antibodies and HIV virions can distribute throughout the entire length of the large intestine in an asymmetric manner. With PET scan guided necropsy, we were also able to locate HIV virions penetrating deeply into the colonic mucosa and in a subset of mesenteric lymph nodes. Taken together, our results show that dIgA antibodies affect viral dissemination and penetration throughout the gut mucosa, potentially through the formation of immune complexes. Our data also validate our combined PET imaging and fluorescent microscopy methodology as a highly efficient approach for non-invasively tracking HIV and antibodies.

## Results

### PET provides a rapid and spatially comprehensive screen for tracking radiolabeled dIgA and reveals dIgA distribution throughout colon and to mesenteric lymph nodes two hours after rectal challenge

To determine whether PET could be used to track dIgA and HIV following rectal challenge, we performed two proof-of-principle experiments: the first one tracked antibody and the second tracked both antibody and virus. In the first, double-tagged dIgA-Cy5-[64]Cu (either dIgA1 or dIgA2) was administered rectally to two RMs, followed 30 minutes later with rectal viral challenge with PA-GFP-BaL (protocol 1 in **Table 1**). Animals were sacrificed at two hours post-viral challenge, and rectum and descending colon removed in one piece. PET imaging revealed dIgA-Cy5-[64]Cu (both dIgA1 and dIgA2) throughout the rectum and descending colon though not uniformly, but in foci (**Fig 1A**). The mesenteric lymph nodes, which were too small to be resolved with whole-body PET, were individually removed and the colorectal tissue cut into pieces. All tissue was frozen and then PET scanned to identify tissue blocks with moderate to high [64]Cu signal, as represented in Fig 1B; these the tissue blocks were used for cryosectioning and further analyses. We were surprised to detect PET signals in some blocks of mesenteric lymph nodes from both dIgA1- and dIgA2-treated animals. The selected blocks were cryosectioned to yield two to three tissue sections (**Fig 1B and S1 Table**), which were then fixed, stained for nuclei, and imaged by deconvolution fluorescent microscopy. Subsequent analysis of the fluorescent images revealed dIgA (i.e. Cy5 signal) in the descending colon (**Fig 1C**) and mesenteric lymph nodes (**Fig 1D**), thereby validating the PET data. The images further revealed that, within the colorectal tissue, the vast majority of dIgA was in the lamina propria, where it appeared to be associated with cells; the phenotype of these cells is currently unknown (**Fig 1C**). Within the mesenteric lymph nodes, dIgA was detected in nodes and also in the subcapsular space (**Fig 1D**). These data clearly demonstrate that dIgA-Cy5-[64]Cu had penetrated and traversed the epithelium into the underlying lamina propria and mesenteric lymph nodes two hours post-rectal challenge.

**Table 1. Identification of rhesus macaques used in each experimental protocol.**

| | | Experimental groups | | | | | |
|---|---|---|---|---|---|---|---|
| | Tissues collected | Animal code | Figures | Antibody | Virus | Virus production | Institute (PET/MRI or PET/CT) |
| Protocol 1 | Rectum, descending colon, mesenteric lymph nodes | 34731 | 1 | dIgA1-Cy5-$^{64}$Cu | PA-GFP-BaL | 293T | RII (PET/MRI) |
| | | 34740 | 1A | dIgA2-Cy5-$^{64}$Cu | | | |
| Protocol 2 | Rectum, descending colon, mesenteric lymph nodes | 35388 | not shown | NA | PA-GFP-BaL-$^{64}$Cu | 293T | RII (PET/MRI) |
| | | 35361 | 1E,F | | | | |
| Protocol 3 | Rectum, descending colon, transverse colon, mesenteric lymph nodes | 34692 | 3–5 | PBS control | PA-GFP-BaL + PA-GFP-BaL-$^{64}$Cu | 293T | RII (PET/MRI) |
| | | 31439 | 4,5B | | | | |
| | | A15X039 | | | | | NIRC (PET/CT) |
| | | A15X085 | 2,4 | | | | |
| | | A15T001 | 4,5B | | | | |
| | | 03D127 | | | | PBMC | |
| | | 34947 | | dIgA1-Cy5 | | 293T | RII (PET/MRI) |
| | | 34711 | | | | | |
| | | 31144 | 4,5B | | | | |
| | | A15X053 | | | | | NIRC (PET/CT) |
| | | A15T006 | | | | | |
| | | A7L008 | 3–5 | | | PBMC | |
| | | 34912 | 3–5 | dIgA2-Cy5 | | 293T | RII (PET/MRI) |
| | | 34362 | | | | | |
| | | A15X065 | | | | | NIRC (PET/CT) |
| | | A19X090 | 4,5B | | | | |
| | | A15X082 | | | | PBMC | |
| | | A15X020 | | | | | |

NIRC, New Iberia Research Center; PBMC, peripheral blood mononuclear cells; RII, Research imaging institute at University of Texas Health Science Center at San Antonio

### PET provides a rapid and spatially comprehensive screen for tracking radiolabeled virus and reveals HIV in colon and mesenteric lymph nodes two hours after rectal challenge

To determine whether PET can be used to track the virus *in vivo*, we performed a second proof-of-principle experiment consisting of rectal viral challenges on two animals with $^{64}$Cu-labeled virus (PA-GFP-BaL-$^{64}$Cu). The virus was first conjugated with a dodecane tetraacetic acid (DOTA) chelator prior to complexing the radioactive copper (see Methods). This process involved multiple incubation steps at varying temperatures, resulting in a loss of infectivity in the PA-GFP-BaL-$^{64}$Cu. Although infectivity was decreased more than 90%, examination of virion associated photoactivatable GFP revealed the particles were still intact. To compensate for the loss of infectivity, we combined the PA-GFP-BaL-$^{64}$Cu, with an excess of PA-GFP-BaL and evaluated the ability of PET scan guided necropsy to identify sites of PA-GFP-BaL localization as revealed by fluorescence microscopy of tissue cryosections.

Initial pilot PET scans of radiolabeled virus were acquired at one and two hours immediately following rectal viral challenge with the mixture of PA-GFP-BaL-$^{64}$Cu, with an excess of PA-GFP-BaL. These scans revealed extensive and asymmetric distribution by the 1-hour time-point with further distribution revealed in the two-hour scan. The PET images revealed $^{64}$Cu signals in the rectum and throughout the descending and transverse colon in a generally

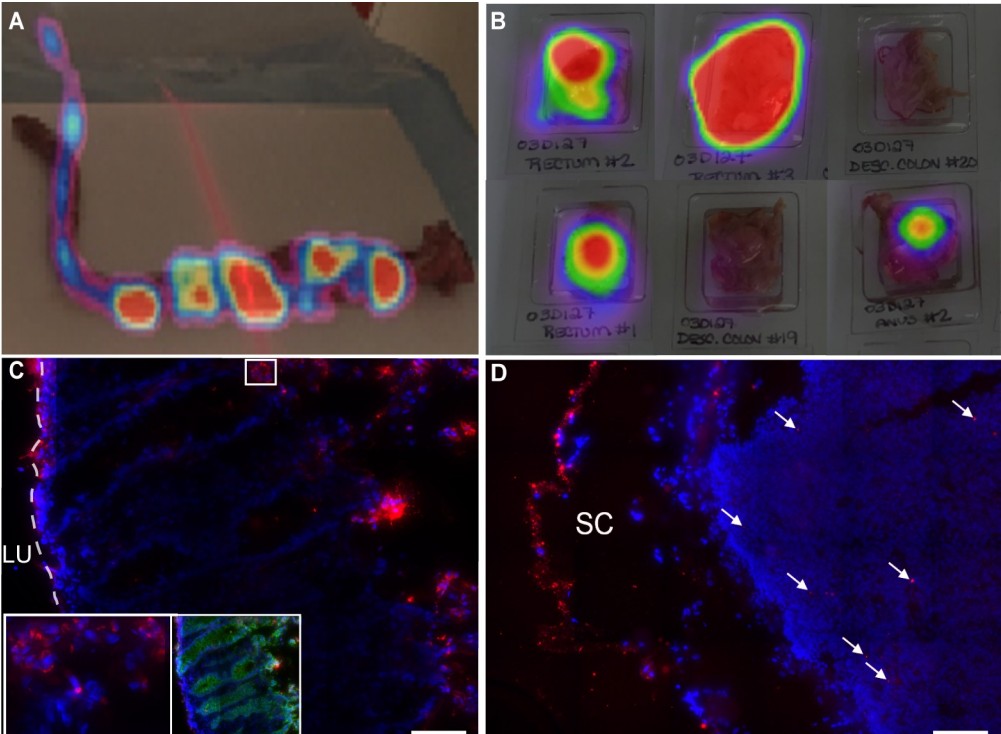

**Fig 1. dIgA penetrates colon epithelium and mesenteric lymph nodes by two hours post rectal challenge.** Two animals were treated rectally with either dIgA1-Cy5-$^{64}$Cu or dIgA2-Cy5- $^{64}$Cu, followed by challenge with PA-GFP-BaL (Table 1, protocol 1) After two hours, animals were sacrificed, the rectum and descending colon removed in one piece, the attached mesenteric lymph nodes separated, and the rectal and colonic tissue cut into pieces. All tissue was then frozen, cryosectioned and processed for fluorescence microscopy. Representative PET images of $^{64}$Cu signal (**A,B**) and fluorescent microscopy images of Cy5 antibodies (**C,D**). (**A**) PET image of dIgA-C45-$^{64}$Cu overlayed on photograph of whole, excised rectum and descending colon. (**B**) PET signal in six blocks of descending colon tissue from the dIgA1 animal (**C**) Cryosection of block of descending colon immunolabeled for E-cadherin (green, right insert). Left inset, enlargement of white boxed area in main image. (**D**) Cryosection of a single mesenteric lymph node. White arrows indicate areas of Cy5 detection (IgA) in the mesenteric lymph node. For **C** and **D**: Red, Cy5 signal; blue, Hoescht stain; LU, lumen; SC, subcapsular space. Scale bars, 40 µm.

decreasing gradient with minimal signal towards the proximal colon (**Fig 2**). After the two-hour PET scan, animals were necropsied, and the rectum and colon were excised in one piece and reanalyzed in an additional PET scan to reveal anatomic details of signal distribution. Tissue was processed to isolate small pieces of tissue that contained robust $^{64}$Cu signal and frozen into blocks for microscopy. The frozen tissue pieces in cryomolds were PET scanned again to identify the individual blocks with maximal $^{64}$Cu signal (**Fig 2D**). Upon imaging of cryosections from the tissue fragments enriched for $^{64}$Cu signal, we readily identified PA-GFP-BaL particles within the tissue blocks with robust $^{64}$Cu signal. (**Fig 2E**). In contrast, PA-GFP-BaL particles were rarely found in tissue blocks with low or no $^{64}$Cu signal. These pilot studies reveal that although the process of modification for $^{64}$Cu labeling of PA-GFP-BaL inactivated the potential for infectivity it did not alter the physical properties of the particles because the $^{64}$Cu labeled PA-GFP-BaL distributed in the colon with the unlabeled infectious virus population. The co-distribution of the radiolabeled and native PA-GFP-BaL populations confirms our ability to use the $^{64}$Cu-labeled virus as a beacon for locating unlabeled virions. These pilot experiments also highlight the need for subsequent studies to consider whether the entire colon needs to be excised as we saw radioactivity past the descending colon.

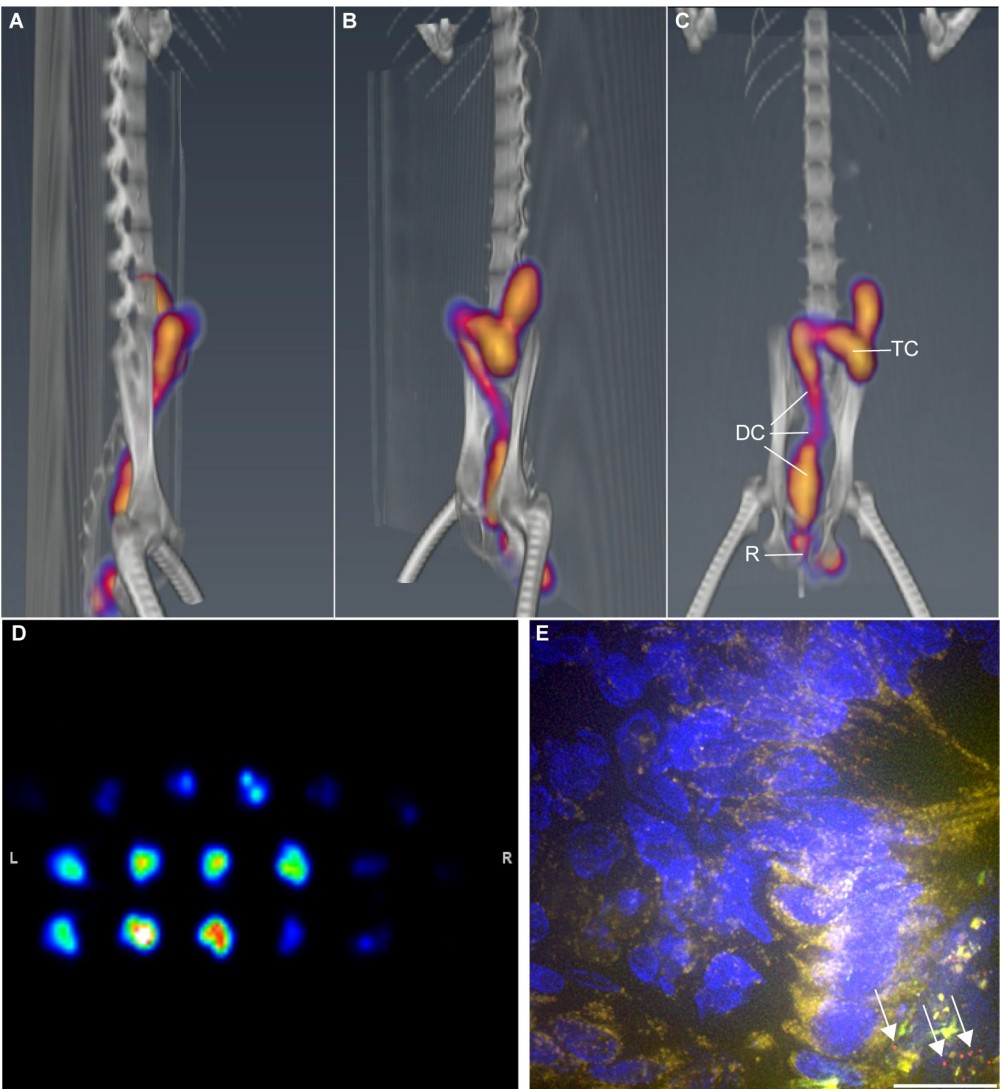

**Fig 2. Virus detected in transverse colon two hours post-rectal challenge.** Animals were challenged rectally with PBS, followed by a mixture of PA-GFP-BaL and PA-GFP-BaL $^{64}$Cu (**Table 1,** protocol 3). Two hours later, the animal was imaged, first by PET/CT, and the images coupled. Representative images showing (**A**) Sagittal, (**B**) diagonal, and (**C**) coronal views. R, rectum; DC, descending colon; TC, transverse colon. (**D**) Representative PET image of colon blocks from animals that were challenged with PA-GFP-BaL $^{64}$Cu. Images were taken after necropsy and tissue processing two hours after rectal viral challenge (**E**) Representative fluorescent microscopy image of cryosection from colon of an animal that was challenged PA GFP-BaL $^{64}$Cu. Red, GFP/HIV virion (post-activation); blue, dIgA; green, autofluorescence (pre-activation). White arrows indicate HIV virions. Scale bars, 20 μm.

With the completion and optimization of our methods and proof-of-principle pilot studies, we then completed an 18-animal study using 3 groups of 6 animals challenged with a 1:2 mixture of PA-GFP-BaL-$^{64}$Cu and PA-GFP-BaL 30 minutes after application of fluorescently tagged antibody (dIgA1-Cy5 or dIgA2-Cy5) or control PBS (**Table 1,** protocol 2). Animals were scanned by whole-body PET at 1 and 2 hours post viral challenge, followed by CT (or MRI; **Table 1**). Coupling of the PET and CT (or MRI) images revealed virus throughout the rectum and descending colon in all 18 animals; signal was also apparent in the transverse colon of 10 of the 18 animals (**Fig 2**). Thus, PET combined with fluorescently-labeled probes can be used to rapidly screen large areas of tissue—*in vivo* or *ex vivo*—for pathogen (or

antibody) that can subsequently be studied microscopically. These novel data further reveal that, in less than two hours, rectal challenge inoculum has asymmetrically distributed in the descending and transverse colon. This extensive and rapid distribution of $^{64}$Cu was confirmed to represent the distribution of PA-GFP-BaL by examination of cryosections. The distribution of radiolabeled virus was more extensive than anticipated but consistent with previous reports of a labeled viral particle simulant ($^{99}$Technetium) that can distribute beyond the splenic flexure in some study participants[30] and distribution of rectally applied India ink[31].

## Rectal application of dIgA influences HIV virion numbers and penetration

To determine the influence of dIgA antibodies on the earliest stages of HIV transmission after rectal challenge, we analyzed virion numbers, penetration, and distribution in the 18 rectally challenged RMs that were administered dIgA1-Cy5, dIgA2-Cy5, or PBS and a mixture of PA-GFP-BaL-$^{64}$Cu and PA-GFP-BaL (**Table 1**, protocol 2). Following whole-body imaging (**Fig 2**), animals were sacrificed, and the rectum, descending colon, and transverse colon were removed as one piece. As described above (Fig 1 Results), mesenteric lymph nodes and 1-cm$^2$ pieces of the colorectal tissue were frozen and PET scanned (as in **Fig 1B**), and the tissue blocks with moderate or high $^{64}$Cu-signal were cryosectioned. The images were subsequently examined for virus and dIgA (**Fig 3**). Virus particles were readily detectable in those blocks of tissues that registered with high radioactivity. However, the number of virions was much lower in transverse colon in all experimental groups, consistent with the decreased $^{64}$Cu signal observed in this part of the colon. The dIgA-Cy5 signal distribution was asymmetric in intensity within individual tissue blocks. No Cy5 signal was detected by fluorescent microscopy in the PBS controls (**Fig 3A**), but Cy5 signal was apparent in rectum, descending, and transverse colon of dIgA1- and dIgA2-challenged animals (**Fig 3B and 3C**). These microscopy data further validate PET as a preliminary screen to localize antibody and virus after rectal challenge.

Next, we quantified data for each of the three colorectal tissues (rectum, descending, and transverse colon) and three experimental groups by counting all PA-GFP-BaL virions associated with mucosa, scoring them as penetrating or not and, for those that were penetrating, measuring the depth of penetration. Penetration was defined as being more than one micron below the epithelial surface. From these data, we generated three outcome measures: total number of virions, proportion of penetrating virions, and the depth of penetration (**Fig 4**; see Methods, Fluorescent microscopy and image analysis, and Statistics). For the first outcome measure (**Fig 4A–4C**), we did not find statistically significant differences in the mean number of virions for the three experimental groups in rectal tissue (**Fig 4A,** PBS, 9.76 ± 6.96; dIgA1, 3.50 ± 2.18; dIgA2, 16.98 ± 8.41) or descending colon (**Fig 4B,** PBS, 15.73 ± 12.05; dIgA1, 8.66 ± 2.84; dIgA2, 36.10 ± 21.47), although the mean values for dIgA2 were consistently larger than dIgA1 or PBS. However, the analysis revealed that the mean number of virions in transverse colon tissue was significantly greater ($P < 0.01$) in animals challenged with dIgA + virus vs PBS + virus ($P < 0.01$ for dIgA1 and dIgA2), and ($P = 0.0164$) in animals challenged with dIgA2 + virus vs dIgA1 + virus (**Fig 4C,** estimated mean number of virions: $P = 0.02$; PBS, 0.008 ± 0.005; dIgA1, 0.29 ± 0.15; dIgA2, 2.24 ± 1.72). These data suggest that the rectal application of dIgA antibodies increases the distribution of HIV virions throughout the colon and that dIgA2 has a greater influence over distribution than dIgA1.

For the second outcome measure (**Fig 4D–4F**), the analysis revealed a significantly higher mean proportion of penetrating virions in rectal tissue from animals challenged with dIgA1 + virus vs PBS + virus ($P < 0.01$), but no difference between dIgA2 + virus vs PBS + virus (**Fig 4D,** PBS, 0.16 ± 0.04; dIgA1, 0.34 ± 0.03, dIgA2, 0.25 ± 0.06). For descending colon tissue, there was a higher mean proportion of penetrating virions in tissue from animals challenged

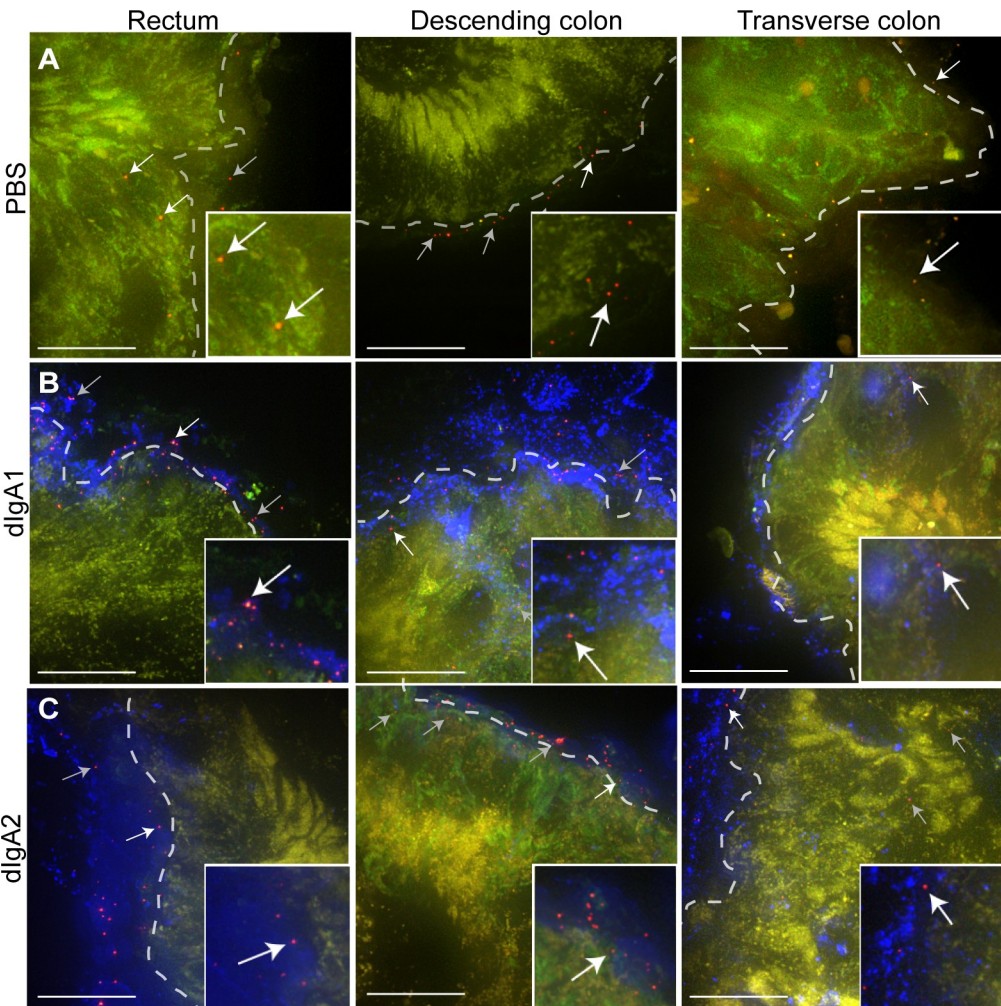

**Fig 3. HIV Virion penetration two hours post-rectal challenge.** Three animals were challenged rectally with PBS, dIgA1-Cy5 or dIgA2-Cy5 and a mixture of PA-GFP-BaL and PA GFP-BaL $^{64}$Cu (**Table 1,** protocol 3). Two hours later, animals were sacrificed, tissues dissected, frozen and tissue blocks with intense PET signal (virion) were cryosectioned. Representative fluorescent microscopy images. Red, GFP/HIV virion (post-activation); blue, dIgA; green, autofluorescence (pre-activation). (**A**) PBS negative control (**B**) dIgA1-Cy5. (**C**) dIgA2-Cy5. White arrows indicate HIV virions shown in insets; grey arrows show other viral particles in micrograph. A dashed line in each panel demarcates the edges of the epithelium. Scale bars, 20 μm.

with PBS + virus vs either dIgA1 or dIgA2 + virus (**Fig 4E,** $P < 0.01$; PBS, 0.23 ±0.04; dIgA1 0.04 ± 0.02; dIgA2, 0.08 ± 0.03). In transverse colon tissue, the analysis did not reveal any statistically significant differences (**Fig 4F,** PBS, 0.22 ± 0.11; dIgA1, 0.07 ± 0.03; dIgA2, 0.06 ± 0.03), however this could be an artifact of the small sample size.

For the third outcome measure (**Fig 4G–4I**), there were no significant differences in mean viral penetration depth between the three treatment groups for tissue from either rectum (**Fig 4G,** PBS, 6.21 ± 0.84; dIgA1, 6.32 ± 0.61; dIgA2, 5.79 ± 0.76) or descending colon (**Fig 4H,** PBS, 5.95 ± 0.66; dIgA1, 7.28 ± 1.07; dIgA2, 5.17 ± 0.36). Mean penetration depth for virions in transverse colon was greater in animals challenged with dIgA2 + virus vs dIgA1 + virus (**Fig 4I**, $P < 0.01$; dIgA1, 6.94 ± 1.03; dIgA2, 10.76 ± 0.005); there was only one penetrating virion in all of the images of transverse colon from animals in the PBS + virus group. Interestingly, the penetrating virions in the transverse colon were associated with dIgA antibodies (**Fig 3**).

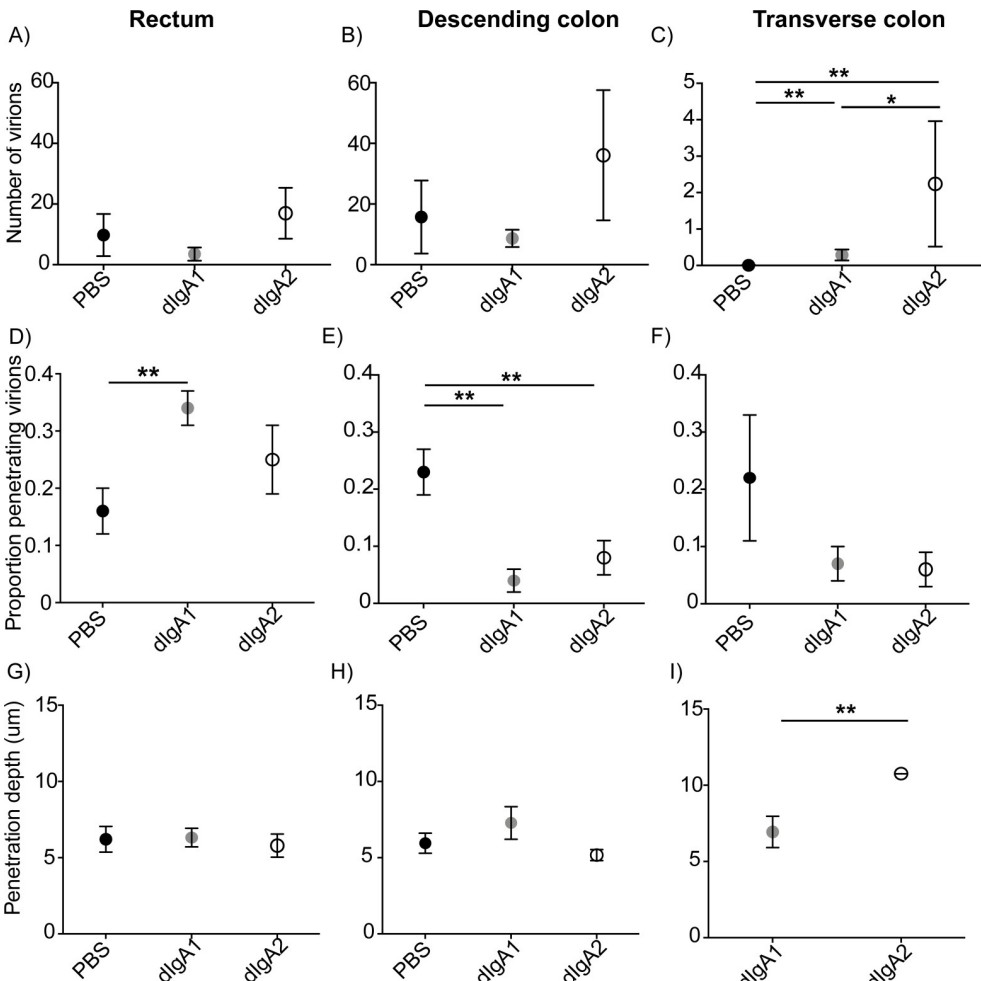

**Fig 4. dIgA antibodies impact virion distribution and penetration after rectal challenge.** Quantification of three outcome measures for the experiment in Fig 3 (see "Outcome Measures" in Methods). (**A-C**) total number of virions counted in images (**D-F**) proportion of penetrating virions (**G-I**) penetration depth. A,D,G: Rect um; B,E,H: Descending colon; C,F,I: Transverse colon. Circles and bars, estimated means and standard errors; n = 6 animals/ group (18 animals total); *, $P < 0.02$; **, $P < 0.01$.

These data suggest that dIgA2 influences the penetration depth of HIV virions into the mucosa.

## HIV is detected in mesenteric lymph nodes as early as two hours after rectal challenge

In our proof-of-principle experiments (**Figs 1 and 2**), we also unexpectedly detected PET ($^{64}$Cu) signals in some of the mesenteric lymph nodes two hours after rectal challenge. The frozen blocks, each containing a single node, were selected and processed for fluorescent microscopy as described above for colorectal tissue. The radiolabeled nodes were confirmed, by fluorescent microscopy, to contain virions (**Fig 5A**). The microscopic analysis also revealed dIgA1 and dIgA2 (i.e. Cy5; **Fig 5A**) in the mesenteric lymph nodes. The fluorescent images were then analyzed as described above for colorectal tissue, except that only outcome measure 1, number of virions, was quantified (**Fig 5B and 5C**). Interestingly, the fluorescent microscopy analysis revealed that while all six dIgA2 + virus animals had detectable virions (PA-GFP

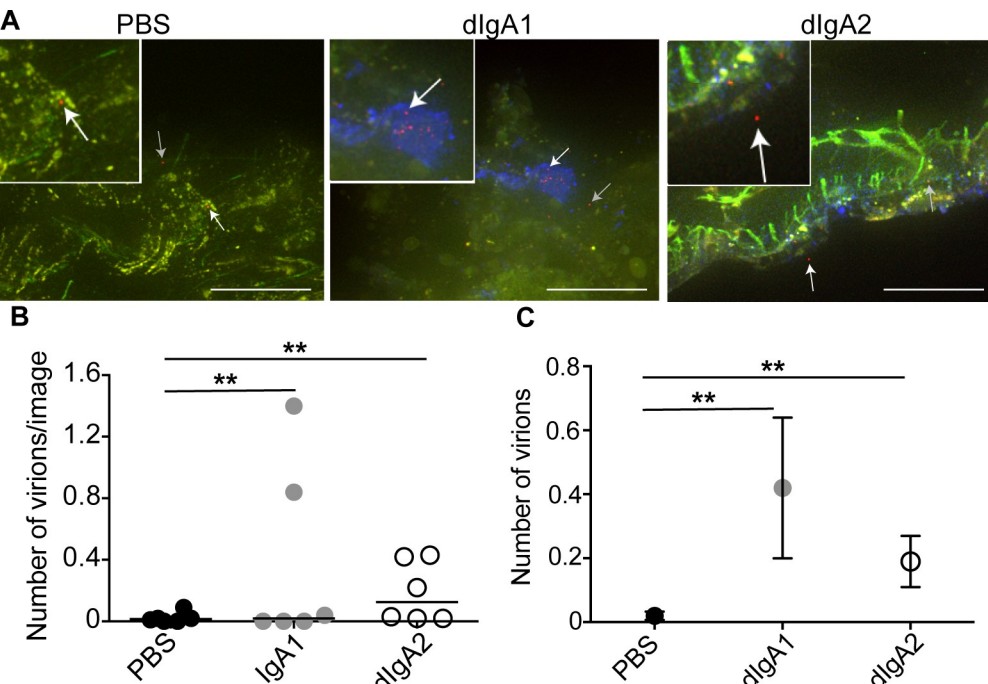

**Fig 5. Number of virions in mesenteric lymph nodes 2 hours after rectal challenge is IgA dependent.** Eighteen animals were challenged rectally with PBS, dIgA1-Cy5 or dIgA2-Cy5 and a mixture of PA-GFP-BaL and PA-GFP-BaL [64]Cu (**Table 1,** protocol 3). Two hours later, animals were sacrificed, mesenteric lymph nodes removed, individually frozen and OCT blocks with moderate to high PET signal cryosectioned. (**A**) Representative fluorescent microscopy images of nodes from each of the three experimental groups (PBS negative control, dIgA1-Cy5 and dIgA2-Cy5). Red, GFP/HIV virion (post-activation); blue, dIgA; green, autofluorescence (pre-activation). White arrows indicate HIV virions in insets; grey arrows show other viral particles in micrograph. Scale bars, 20 μm. (**B-C**) Quantification of total number of virions found in cryosections of mesenteric lymph nodes for the experiment in Fig 3 (**Table 1**, protocol 3). (**B**) Dots depict individual animals and bars depict. (**C**) Circles and bars, estimated means and standard errors; n = 6 animals/group (18 animals total); $^{**}$, $P < 0.01$.

signal) in cryosections from at least some mesenteric lymph nodes; only four animals in the PBS + virus group and three in the dIgA1 + virus group had one or more virions in all the images of mesenteric lymph nodes (**Fig 5B**). Interestingly, images containing virions tended to be from mesenteric lymph nodes removed from the transition zone between descending and transverse colon. The quantitative analysis revealed a statistically greater number of virions in mesenteric lymph nodes from animals challenged with dIgA1 or dIgA2 + virus vs PBS + virus ($P < 0.01$ and $P = 0.006$, respectively; PBS, 0.02 ± 0.013; dIgA1, 0.42 ± 0.22; dIgA2, 0.19 ± 0.08). On the other hand, the difference between dIgA1 and dIgA2 was not statistically significant (**Fig 5C**). Thus, in less than two hours post-challenge, a fraction of the HIV virions (and dIgA) was detected in the mesenteric lymph nodes, with relocation facilitated by dIgA, particularly dIgA2.

## Discussion

Secretory IgA antibody-mediated protection of mucosal tissues from pathogens holds great potential as an HIV vaccine strategy. However, the current generation of injectable vaccines stimulates an immune response that, unfortunately, leads to only weak induction of secretory IgAs[32]. Data from our group has however demonstrated the ability of topically administered anti-HIV IgA to inhibit and even prevent SHIV acquisition[19]. To advance HIV vaccine research it is essential that we can identify the underlying mechanisms of how mucosally

applied dIgA2, combined with i.v. IgG1 provide complete protection together, although neither has much impact to prevent systemic infection when tested individually. In this study, we determine whether rectally applied dIgA affects the distribution and/or penetration of HIV in an NHP rectal challenge model, utilizing a novel combination of PET imaging and fluorescent microscopy to track the dIgA antibodies and HIV virions. PET allowed us to rapidly screen the entire living animal at multiple timepoints to provide an unbiased survey of the distribution and localization of radiolabeled antibody under the specific conditions of the foundational experiments. This approach allowed us to localize sites of virus or antibody accumulation by examining the complete and intact colorectal mucosa and associated mesenteric lymph nodes for moderate to high $^{64}$Cu signal associated with either dIgA (dIgA-Cy5-$^{64}$Cu) or virus (PA-GFP-BaL-$^{64}$Cu). Once localized, we were able to isolate and cryosection the areas of interest for fluorescent microscopy analysis of Cy5 or photoactivatable GFP, respectively. Compared to similar studies in the female reproductive tract that did not use PET and double-labeled ($^{64}$Cu and GFP) virus [23,33], our rapid PET screen dramatically increased the probability of finding virions (~3,000 virions vs ~73,000 virions) in an unbiased approach, which, in turn, allowed for robust statistical analysis of most outcome measures. It also dramatically increased the amount of tissue that could be analyzed and evaluated.

For these studies, we utilized a PA-GFP labeled replication competent HIV-BaL which has important implications. Utilizing PA-GFP-VPR allows robust labeling and detection of individual particles that are efficiently bound by the various HGN194 isotypes in our study. We are not currently able to efficiently label SIV Gag based particles with tagged SIV VPR. The PA-GFP-BaL cannot infect potential target cells in the RM model because of host restriction factors such as TRIM5a [34,35]. The use of PA-GFP-BaL also allows us to compare outcomes to other studies defining virion distribution in NHP and explant culture models. It is also notable that the amount of virus in this model is super-physiological compared to what is speculated for human transmission [23]. Large amounts of viral particles are needed for this efficient tracking. However, the antibody concentrations are also super-physiological, as is the single, large dose challenge approach used to identify the synergistic blocking activity of HGN194 dIgA2 and HGN194 IgG1 to block SHIV-1157-ipEL-p [19].

Our studies demonstrate that the distribution of dIgA antibodies and virus, only two hours after rectal challenge,[36–38] was much broader than expected, based on previous studies by our lab and others that employed SPECT imaging in the macaques [31] and humans [39]. Virus and dIgA antibodies were found in the rectum and also in the descending and transverse colon. In all three regions, we determined that much of the dIgA and virions had penetrated the epithelium and were detected in the underlying lamina propria of the mucosa, where they often appeared to be cell-associated. Future experiments will aim to determine the phenotype of these cells. Most surprising, perhaps, was that virions had spread beyond the mucosal tissue and were detected along with dIgA in mesenteric lymph nodes, which are thought to facilitate viral spread due to the large number of immune cells present[40]. Our results show that the majority of virions in animals given dIgA were associated with antibodies (**Fig 3**). While we hypothesize that the virions and antibodies that were observed in mesenteric lymph nodes are in immune complexes, further experiments are required to ascertain this.

Our data on the effects of rectal application of either HGN194 dIgA1 or dIgA2 (vs PBS negative control) on viral distribution revealed statistically significant differences two hours after challenge. Unexpectedly, the presence of either HGN194 dIgA1 or dIgA2 led to a greater distribution of the viral particles. Animals challenged with dIgA plus virus had greater numbers of virions in transverse colon and mesenteric lymph nodes, as well as an increased probability of having penetrating virions in the rectum. These results suggest underlying transport mechanisms for dIgA1 or dIgA2 immune complexes that impact their distribution within the

mucosal milieu. In contrast to the increased probability of dIgA1/2-HIV immune complexes to penetrate the rectal tissue, we found fewer penetrating virions in the descending colon of animals given dIgA compared to PBS controls. This finding could be due to differences in the mucosal barriers (tissue structure, mucus composition, etc.) between the descending colon and the rectum. Such differences have been recently reported in mice [41]. Alternatively, the dIgA stimulated trafficking to the mesenteric lymph nodes that drain the descending colon in animals given dIgA would move penetrating virions deeper into the tissue and beyond our analysis of the epithelial barriers.

Finally, we noted some differences between dIgA1 and dIgA2. In transverse colon, dIgA2-treated animals had significantly more virions and their depth of penetration was significantly greater. This observation suggests that dIgA2 immune complexes facilitate a wider general distribution of immune complexes through the colon. This is consistent with our observation that the mean number of virions was also greater in rectum and descending colon from dIgA2-treated animals, although these differences were not significant. In humans, IgA1 is predominantly found in nasal and oral cavities, and well as the bronchial mucosa and small intestine, while IgA2 is predominately found in the distal small intestine and the colon.[4,5] This difference in IgA subclass distribution could also potentially explain the differences we observed in the transverse colon between animals given dIgA1 and dIgA2. Our data suggest that the immune complexes of the virus and HGN194 dIgA2 are more mobile relative to similar immune complexes formed with HGN194 dIgA1. This interpretation is supported by our observation that the viral particles appear more efficiently in the mesenteric lymph nodes and more abundantly in the transverse colon in the presence of HGN194 dIgA2 compared to HGN194 dIgA1. This increase in mobility could be facilitated by HGN194 dIgA2 engagement of virus in the rectal environment or through interactions with receptors on resident cells.

These data demonstrate that the radius of distribution for virions at two hours post-challenge was much greater than previously thought, reaching transverse colon and mesenteric lymph nodes. The data also demonstrate that rectal application of dIgA antibodies 30 minutes before HIV challenge increases the number of virions in these distal locations and that these effects are greater for dIgA2, except in the rectum. The greater number of virions in dIgA2 might be explained by the fact that the hinge region of IgA1 increases its sensitivity to bacterial proteases, which may explain why IgA2 is found in greater abundance at gut mucosal sites [2,42], and may allow IgA2 to form more stable immune complexes with PA-GFP-BaL. These data support a model in which dIgA antibodies form immune complexes with HIV in the gut mucosa to shuttle the virions to mesenteric lymph nodes. Evidence from previous experiments suggests that the dIgA-HIV immune complexes are shuttled by cells[6,8,9]. Future studies will determine the cell types that shuttle the immune complexes, and the mechanism that may be responsible for the trafficking of dIgA antibodies and virions to the mesenteric lymph nodes after viral rectal challenge. It is notable that a previous manuscript currently describes similar observations of orally instilled murine leukemia virus being rapidly transported to the mesenteric lymph nodes, indicating conserved pathways associated with immune surveillance of the gut mucosa and underlying mucosal immunology[43].

The RV144 trial showed increased binding of Env-specific IgA in the plasma directly correlated to infection in the study participants,[44] fostering the idea that IgA enhanced viral infection. One potential paradox of the observations here is that dIgA1 and dIgA2 facilitate the delivery of viral particles to the draining mesenteric lymph nodes. This might be anticipated to actually increase viral acquisition which is not concurrent with our previous observation that HGN194 dIgA1 protected 5/6 animals while HGN194 dIgA2 protected 1/6 animals in a single high dose rectal challenge model.[19] A key consideration in relating the observations reported here to the protective challenge studies, is that the IgG1 required for complete protection was

not included in the experiment. Future studies will determine the mechanistic impact of the i. v. IgG1 in the 100% protection observed with the combination of HGN194 dIgA2 with HGN194 IgG1. The issue of increased potential transmission of virus transported to the mesenteric lymph nodes in immune complexes would be moot, if the virions within the immune complexes are inactivated or neutralized. However, the conditions of the original study, with a single high dose challenge do not allow the detection of any potential increase in transmission efficiency. It is interesting to consider that this inactivation in immune complexes might be different for dIgA1 and dIgA2 explaining why HGN194 dIgA1 provides potent protection (5/6) while HGN194 dIgA2 only provides minor protection (1/6 protected). Future studies will determine how HGN194 IgG1 alters the milieu to provide 100% protection and the possibility that dIgA mediated delivery of virus complexes to the mesenteric lymph nodes can lead to systemic infection. These observations could also lead to a hypothesis that vaccine induced dIgAs that are suboptimal at, if not outright unable to minimize virus infectivity may actually enhance virus acquisition via these disseminating mechanisms, a hypothesis that needs to be investigated during future HIV vaccine trials.

Based on the initial observations reported here, we hypothesize that the observed synergy of inhibition observed by the combination of two inefficient antibodies (HGN194 IgG1 and HGN194 dIgA2) has a 2-step mechanism. First, interaction with HGN194 dIgA2 in the mucosal environment traps many particles by immune exclusion reducing the challenge dose. The inefficient protection by HGN194 IgG1 revealed by an observed delay in peak virus load after high dose challenge is sufficient to prevent systemic infection by the small virus challenge dose avoiding immune exclusion. Future studies described above will provide insights into the underlying mechanism of protection.

The substantial data generated in this study lend strong support to our hypothesis that induction of HIV-specific secretory IgA containing dIgA is a promising strategy for HIV vaccine development. This comprehensive analysis was made possible by our innovative two-step PET-fluorescent microscopy imaging of double-labeled virus (or dIgA). With this method, as we have clearly demonstrated, PET can be used as an *in vivo* screen to pinpoint the locations of a radiolabeled probe, and then those select areas can be rapidly selected and subjected to microscopic and/or molecular analyses by exploiting a fluorescent label on the same probe.

## Methods

### Ethics statement

All primate studies (including PET, MRI, and CT imaging, below) were conducted at the Research Imaging Institute at UT Health San Antonio (RII/UTHSA) or at the New Iberia Research Center (NIRC) at the University of Louisiana at Lafayette (**Table 1**). All procedures were approved by the Animal Care and Use Committees of both UTHSA (ACUC: 20160 070AR) and the University of Louisiana at Lafayette (ACUC:2017-8789-005). All studies were performed in accordance with the recommendations in the Guide for the Care and Use of Laboratory Animals.

### Virus production

To generate PA-GFP-BaL, we co-transfected the R5-tropic, R9-BaL infectious molecular clone construct with a plasmid expressing a photoactivable GFP (PA-GFP) [29] fusion with HIV VPR (PA-GFP-VPR) as previously described[22,23]). The replication competent virus labeled with PA-GFP-VPR generated by polyethylenimine transfection of human 293T cells in DMEM medium containing 10% heat-inactivated fetal calf serum, 100 U/ml penicillin, 100 µg/ml streptomycin, and 2 mM l-glutamine. After 24 to 48 hours, virus was harvested,

filtered at 0.45 μm and stored at -80˚C [23]. We also produced PA-GFP-BaL in human peripheral blood mononuclear cells (PBMCs)—a more biologically relevant model; this viral stock, which was used in 4 of the 20 animals (**Table 1**), was purified using magnetic-activated cell sorting to deplete the supernatant of CD45+ cellular debris (Miltenyi Biotec, Bergisch Gladbach, Germany)[45,46]. The particles were concentrated and enriched by centrifugation through a sucrose cushion.

## DOTA labeling of virus

PA-GFP-BaL was labeled with a dodecane tetraacetic acid (DOTA) chelator, which allowed for attachment of $^{64}$Cu. Two buffers were prepared using a chelating resin to remove all free copper: 0.1 M sodium phosphate buffer (pH 7.3) and 0.1 M ammonium acetate buffer (pH 5.5). Chelex 100 Chelating Resin (5 g, BioRad, Hercules, California) was added to 100 ml of each buffer, incubated with stirring for 1 hour at room temperature, and sterilized by filtration at 0.22 μm. Concentrated virus was resuspended in PBS and a 1:10 volume of 1 M sodium bicarbonate added. DOTA-NHS-ester (Macrocyclics, Dallas, Texas) was dissolved in the 0.1 M sodium phosphate buffer. The two solutions were combined (0.3 mg DOTA-NHS-ester per 500 ng of virus, as detected by p24 assay), and incubated at room temperature on a rocker in the dark. After 30 minutes, the buffer was exchanged for the 0.1 M ammonium acetate using a Zeba column 40K (Thermo Fisher Scientific, Waltham, Massachusetts), wash steps completed per manufacturer's protocol, and virus (PA-GFP-BaL-$^{64}$Cu) collected and frozen for shipment to Research Imaging Institute at UT Health San Antonio (RII-UTHSA) or New Iberia Research Center (NIRC) at the University of Louisiana at Lafayette.

## $^{64}$Cu labeling of virus particles

A solution of $^{64}$CuCl$_2$ (University of Wisconsin-Madison or Washington University, St. Louis, MO) was neutralized with Chelex-treated 1 M NH$_4$OAc (Sigma, St. Louis, Missouri) to a pH of 5.5, and an aliquot (~185 MBq) incubated with DOTA-PA-GFP-BaL stock for 1 hour at 37˚C. The sample was purified with a Zeba desalting spin column (40K MWCO, Thermo Fisher), eluted with PBS (Thermo Fisher), and the purity of each dose determined by instant thin-layer chromatography. Labeled virus (10–37 MBq) was mixed with unlabeled virus immediately prior to intrarectal application.

## Cy5 labeling of antibody

20mgs of HGN194-dIgA1 (dIgA1) or HGN194-dIgA2 (dIgA2) were combined with 0.5 mg Cy5 fluorophore (sulfocyanine 5 NHS ester, Lumiprobe, Hunt Valley, Maryland) in 10 mL PBS with 100 mM sodium bicarbonate (as we previously described[47]), and gently rocked at room temperature in the dark. After 1 hour, solutions were passed through two 10-mL Zebra columns and buffer exchanged with PBS to remove free dye. Cy5-labeled antibodies were passed through a 0.45-μm filter and stored at 4˚C protected from light.

## NHP studies: animal care, rectal antibody applications, and viral challenge

In total, 20 Indian-origin RMs (*Macaca mulatta*) were used—two to generate the proof-of-principle data in Fig 1 (**Table 1**, protocol 1), and 18 for the data in all other figures (**Table 1**, protocol 3). For protocol 1, double-tagged antibody (either dIgA1-Cy5-$^{64}$Cu or dIgA2-Cy5-$^{64}$Cu) was rectally applied and, 30 minutes later, both animals were rectally challenged with PA-GFP-BaL (1.5 mL at 1000 ng/ml), as previously described [20]. For protocol 2, PA-GFP-BaL-$^{64}$Cu was rectally applied (1.5 mL at 1000 ng/ml). For protocol 3, the 18 RMs

were divided into three experimental groups (six animals per group), receiving rectal application of PBS, dIgA1-Cy5, or dIgA2-Cy5 (1.25 mg in 0.5 ml). After 30 minutes, animals were rectally challenged with a mixture of 0.5 mL PA-GFP-BaL-$^{64}$Cu and 1 mL unlabeled PA-GFP-BaL (1000 ng/ml). Animals were humanely sacrificed with an overdose (100 mg/kg) of pentobarbital while under isoflurane anesthesia (Euthasol, Virbac, Westlake, Texas) or telazol anesthesia. Colorectal tissue was removed as one piece (rectum, descending colon, and transverse colon for protocol 2; rectum and descending colon for protocol 1). Mesenteric lymph nodes were then individually removed, and the colorectal tissue cut into 1-cm$^2$ pieces. Lymph nodes and colorectal tissue pieces were frozen in optimal cutting temperature (OCT) media (Thermo Fisher Scientific). Once the frozen tissue was no longer radioactive, it was shipped on dry ice to the Hope Lab at Northwestern University.

## NHP studies: imaging (PET, CT and MRI)

All PET, CT, and MRI imaging was performed at either RII/UTHSA (only MRI) or NIRC (only CT). Animals were sedated with 10 mg/kg intramuscularly with ketamine and 1% to 3% isoflurane in 100% oxygen at RII/UTHSA or with Telazol/ketamine (Zoeis, Parsippany-Troy Hills, New Jersey) at NIRC. The animal's body was immobilized in dorsal recumbency in a vacuum-sealed veterinary positioner, and body temperature maintained with a warm air blanket (3M Bair hugger Model 505 warming unit, Saint Paul, Minnesota) and water-circulating heating pads. Physiological parameters were continuously monitored: end-tidal $P_{CO2}$, electrocardiogram, heart rate, and respiratory rate (3T MRI Scanner). In addition, respiration, movement, and mucosal coloration (PET Scanner) were visually assessed. Scans lasted 20 minutes.

All 20 animals (Table 1) were PET scanned five times; the first three times were wholebody scans obtained immediately following rectal viral challenge, and at one and two hours post-challenge; of the three whole-body PET scans, only the third is shown (Fig 2). Immediately following this third PET scan, whole-body scans—either CT (NIRC) or MRI (RII-UTHSA) were performed. Animals were then sacrificed, and colorectal tissue removed as one piece and PET scanned. The fifth and last PET scan was of the frozen blocks of colorectal tissue and mesenteric lymph nodes (see "NHP studies," above), and these final images were compared to a standard positive control with "high" signal (0.5mCi, Fig 1B). Tissue blocks with $^{64}$Cu signal equal to or greater than the standard were selected for cryosectioning and further analysis. In a few rare instances, an animal had insufficient numbers of high signal blocks for a given tissue type; in these cases, tissue blocks with moderate signal were used as well.

At RII-UTHSA PET images were acquired using an EXACT HR+ scanner (CTI PET Systems, Knoxville, Tennessee) in 3D mode in a 15.5-cm axial field of view with 63 2.5-mm contiguous slices. Emission data were corrected for decay, dead time, scatter, random coincidences, and measured photon attenuation (with $^{68}$Ge/$^{68}$Ga transmission scans) using scanner software (ECAT v7.2; CTI PET Systems). Corrected image data were reconstructed using OSEM with four iterations and 16 subsets, applying a matrix size of 256×256 and a 5-mm FWHM standard Gaussian filter. All acquired image data were archived on an XNAT-powered data archival system for later analysis. At NIRC, PET/CT scans were acquired using a Philips Gemini TF64 PET/CT scanner. The final CT image was compiled from 250 to 300 slices, depending on animal size. PET-CT combined images were analyzed using OsiriX software. Standard uptake values were measured using the volume regions of interest (ROI) tool and compared and normalized across animals.

Standard 3D anatomical MRI scans were acquired on a 3.0T TIM-Trio MRI scanner (Siemens Healthcare, Erlangen, Germany) with a large 6-channel body-matrix and 12-channel spine-matrix phased-array coils. Upper and lower body 3D images were acquired separately to

cover the whole length of the animal, with about 30% overlap between the two image sets. Two to three common fiducial markers were included in each set to assist the PET/MRI co-registration and image merging. The structural T1 images were acquired with a 3D Flash sequence (6.33/1.51-ms repetition/echo time; 10˚ flip angle; 512×512 matrix; 500x250 mm coronal field of view; 128 1.2-mm slices, without gap; 6 averages; accelerate factor 2 GRAPPA, and 366-s scan time). PET and MRI data analyses and PET/MRI co-registrations were performed using Multi-Image Analysis GUI (MANGO, Research Imaging Institute, UT Health San Antonio). ROIs were drawn on target tissues by placing spherical outlines, free-hand outline drawing, and/or by signal thresholding of tissue contours on the PET image.

### Fluorescent microscopy and image analysis

Blocks of frozen tissue with moderate-to-high PET signal were partially sectioned (10–12 μm thickness), to generate only two to three sections per block; when an animal/tissue did not yield any high signal blocks, "moderate" signal blocks were sectioned instead. Sections were fixed in 3.7% formaldehyde in PIPES buffer for 15 minutes at room temperature, washed in PBS, and Hoescht stained (1:25,000, ThermoFisher Scientific) for 15 minutes. Coverslips were mounted with Dako Cytomation mounting medium (Burlington, Ontario, Canada) and sealed with clear nail polish. Images were obtained by deconvolution microscopy on a DeltaVision inverted microscope (GE, Boston, Massachusetts).

For image quantitative studies (**Figs 3–5**), a 100x lens was used to take 20–22 Z-stack images at 0.5-μm steps of the four tissues (rectum, descending colon, transverse colon, and mesenteric lymph nodes) (**S1 Table**). The 20–22 images were taken in an unbiased manner, by simply following the linear path of the luminal surface of the mucosa—with the distance between images dictated by the bleaching caused by the photoactivation. Images were later analyzed manually using softWoRx software (Applied Precision, Issaquah, Washington) and methods that are well established in the Hope laboratory[22,23] to derive three outcome measures of virions: 1) total number of virions visible in the images, 2) total number of virions penetrating the epithelium, and 3) the depth of penetration. For all three measures, the full area of each of the 20–22 images was analyzed. Penetrating virions were defined as being at least 1 μm beneath the epithelial surface; the probability of a virion penetrating the epithelium, if at least one virion was present in that image, was derived from outcome measures one and two.

### Statistics

In the first set of analyses, we tested whether there were differences between the three rectal-challenge groups (PBS+virus, dIgA1+virus and dIgA2+virus); this was done for each of the three outcome measures (described above) and each of the three colorectal tissue types (rectum, descending colon, and transverse colon). For the mesenteric lymph nodes, only outcome measure 1 was examined. Note, we used generalized estimating equations (GEEs) to account for multiple observations per monkey and to provide model estimated means and standard errors (SEs). For outcome measure 1, the unit of observation was the count per image (with 20–22 images per block, 88–110 images per tissue type or 352–440 images per RM; **S1 Table**). Zero inflated negative binomial GEE models were run to test whether the number of virions was different for the three rectal-challenge groups stratified by tissue type. For outcome measure 2, the unit of observation was for each virion (including images with at least one virion present), and we used binomial GEEs to evaluate if there were differences in the estimated mean probability of a virion penetrating tissue for the three rectal-challenge groups stratified by tissue type. For outcome measures 3, the unit of observation was for each virion that penetrated the tissue, and we used Gamma GEEs to determine if there were differences in the

depth of the virions for the three rectal-challenge groups stratified by tissue type. Post-hoc pairwise tests were performed to test which tissues were different using a Bonferroni corrected alpha of 0.0166. The only exception was the third outcome measure (penetration depth) for PBS+virus-treated transverse colon, where there was only one virion in the images—not enough to include that treatment group. Therefore, for transverse colon, we tested virus+IgA1 and virus+IgA2 using a significance level of 0.05.

## Supporting information

**S1 Table. Number of tissue blocks and images per rhesus macaque in Protocol 3.** * 2–3 sections were placed on 1 slide; a second 2–3 section slide was created as a reserve/back up. ** since mesenteric lymph nodes are small, relative to the large 1 $cm^2$ sections of colorectal tissue, often 2 sections per block were needed to get 22 images. *** Using a 100x lens, the 22 images (per section) were taken by simply following the linear path of the luminal surface of the colorectal mucosa (distance between images was dictated by the bleaching caused by photoactivation). Images were shot and then only later examined and analyzed.
(PDF)

## Acknowledgments

Thanks to Antonio Lanzavecchia, Davide Corti, and Elisabetta Cameroni at Humabs, a subsidiary of Vir Biotechnology, for the production of HGN194 antibodies and to the research and animal care teams at RII/UTHSA and NIRC.

## Author Contributions

**Conceptualization:** Thomas J. Hope.

**Data curation:** Roslyn A. Taylor, Sixia Xiao, Ann M. Carias, Michael D. McRaven, Divya N. Thakkar, Mariluz Araínga, Edward J. Allen, Sidath C. Kumarapperuma, Siqi Gong, Meegan R. Anderson, Yanique Thomas, Beth Goins.

**Formal analysis:** Roslyn A. Taylor, Angela J. Fought.

**Funding acquisition:** Sidath C. Kumarapperuma, Beth Goins, Peter Fox, Francois J. Villinger, Ruth M. Ruprecht, Thomas J. Hope.

**Investigation:** Roslyn A. Taylor, Sixia Xiao, Ann M. Carias, Divya N. Thakkar.

**Methodology:** Roslyn A. Taylor, Ann M. Carias, Michael D. McRaven, Mariluz Araínga, Edward J. Allen, Kenneth A. Rogers, Sidath C. Kumarapperuma, Siqi Gong, Jeffrey R. Schneider.

**Project administration:** Ruth M. Ruprecht, Thomas J. Hope.

**Resources:** Sidath C. Kumarapperuma, Beth Goins, Peter Fox, Francois J. Villinger, Ruth M. Ruprecht, Thomas J. Hope.

**Supervision:** Sidath C. Kumarapperuma, Beth Goins, Peter Fox, Francois J. Villinger, Ruth M. Ruprecht, Thomas J. Hope.

**Validation:** Roslyn A. Taylor.

**Visualization:** Roslyn A. Taylor, Thomas J. Hope.

**Writing – original draft:** Roslyn A. Taylor.

**Writing – review & editing:** Roslyn A. Taylor, Sixia Xiao, Ann M. Carias, Michael D. McRaven, Divya N. Thakkar, Mariluz Araínga, Edward J. Allen, Kenneth A. Rogers, Sidath C. Kumarapperuma, Siqi Gong, Angela J. Fought, Meegan R. Anderson, Yanique Thomas, Jeffrey R. Schneider, Francois J. Villinger, Ruth M. Ruprecht, Thomas J. Hope.

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
