## [Decision Letter · Decision Letter 0]

24 Mar 2021

Dear Professor Hope,

Thank you very much for submitting your manuscript "PET/CT targeted tissue sampling reveals virus specific dIgA can alter the distribution and localization of HIV after rectal exposure." for consideration at PLOS Pathogens. As with all papers reviewed by the journal, your manuscript was reviewed by members of the editorial board and by several independent reviewers. In light of the reviews (below this email), we would like to invite the resubmission of a significantly-revised version that takes into account the reviewers' comments.

We cannot make any decision about publication until we have seen the revised manuscript and your response to the reviewers' comments. Your revised manuscript is also likely to be sent to reviewers for further evaluation.

Sincerely,

Daniel C. Douek

Associate Editor

PLOS Pathogens

Richard Koup

Section Editor

PLOS Pathogens

Kasturi Haldar

Editor-in-Chief

PLOS Pathogens

orcid.org/0000-0001-5065-158X

Michael Malim

Editor-in-Chief

PLOS Pathogens

orcid.org/0000-0002-7699-2064

Reviewer's Responses to Questions

**Part I - Summary**

Reviewer #1: Roslyn Taylor and colleagues used PET and fluorescence imaging to visualize the transmission and dissemination of HIV through the rectal route in non-human primates in the absence and presence of dimeric IgAs. The most surprising finding is how deep viruses penetrate within just two hours. In the presence of IgA viruses, virus entrapment in tissues increases, particles reach deeper and even arrive at the draining mesenteric lymph node. The authors observe interesting differences between IgA1 and IgA2. The hypothesis is that tissue retention and penetration are mediated by immune cells that capture viruses. These features are likely mediated by distinct Fc receptor expressing cells in various tissues.

Technologically the work is very elegant as it combines non-invasive PET with fluorescence imaging with both labeled virus and labeled IgAs. That the virus labeling method leads to reduced infectivity is likely not so important as these initial events are likely dominated by particle transport. This report clearly increases the appetite for more, but given the complexity, cost of the in vivo work, its pioneering nature, I do support publication in PLoS Pathogens. It is also highly relevant for the development of an effective vaccine that provides effective mucosal immunity.

Reviewer #2: This is a very interesting - though preliminary - study of early viral dissemination in the GI tract in the presence of dimeric IgA (dIgA), using combined PET-CT imaging and fluorescence microscopy. The study, performed in rhesus macaques, utilizes Cy5-conjugated dIgA with specificity for HIV Env, and 64Cu-labeled HIV expressing a photoactivatable form of GFP. The virus is therefore both fluorescent and radioactive, and the antibody is fluorescent but distinguishable from the virus. It is worth bearing in mind that HIV replication is severely restricted in rhesus macaques due to host factors, so the system described here has very real limitations and cannot be used to study the impact of virus-antibody interactions on the disease process. As a tool for assessing early viral dissemination without replication, though, the system seems to work well enough to make preliminary observations. The study addresses a limitation of previous virus localization studies that relied on fluorescence microscopy: the detection of fluorescence in situ is limited by tissue opacity, whereas detection of radioactive signal is minimally impacted by tissue thickness.

The findings as described in the text are intriguing and worthy of further study, particularly in light of the previous observation (referenced in the text) that HGN194 dIgA1 protected 5/6 animals from a high dose SHIV challenge. The role of sIgA in sequestering mucosally-acquired virus and perhaps facilitating its transport (as implied here) to MLN, and eventual antigen presentation, is quite intriguing and warrants additional follow-up.

Unfortunately I found several of the Figures unconvincing as currently presented; red virions are quite difficult to spot in these images; perhaps software is more effective than the naked eye.

Reviewer #3: Manuscript by Taylor et al evaluates the ability of PET/CT scanning to evaluate the ability of dIgA to influence the distribution of HIV after rectal exposure. The manuscript takes a highly innovative approach to ask an important question regarding the interplay between the HIV virus and a mucosal antibody at an important mucosal site.

**Part II – Major Issues: Key Experiments Required for Acceptance**

Reviewer #1: (No Response)

Reviewer #2: - HIV replication is restricted in rhesus macaques due to host factors, so the system described here has very real limitations and cannot be used to study the impact of virus-antibody interactions on the disease process, or indeed anything much beyond initial dissemination of inoculum. Some discussion of these limitations (which are not addressed at all in the current text) would be helpful and would help put the study in perspective. In the current draft there is practically no discussion of the virus stock although it is said to be HIV BaL and not SHIV-BaL. (And the Discussion section talks about this study and earlier work, e.g., Ref. 17, without addressing this important difference).

- The virus in these immune complexes is most likely targeted for degradation and/or antigen presentation, particularly if the so-called "mystery" cells are APCs. It would be interesting to use this model to track what happens next (degradation, Ag presentation), for example within the MLN. Do sIgA-mediated immune complexes, and their transport to MLN, enhance presentation of viral antigen, perhaps leading to enhanced mucosal immune responses?

- How the challenge doses compare to the amount of virus that would be "received" during a sexual encounter is not clear.

- As the authors point out, it will also be important in future studies to include IgG1 to determine its mechanistic impact.

- I have looked long and hard at Figure 1D, but I only see one very faint red signal at the end of one of the white arrows; I don't see any others. Can a better image or exposure be found? Also, the images in Figures 3 and 5 are not particularly convincing for red fluorescence, with a few exceptions.

Reviewer #3: While the study was well designed and the technology is impressive, it is not clear that the data presented fully backs up the conclusions that are reached, this is in part due to the difficulties associated with the microscopic images. The issues:

1. Microscopy images are from frozen tissue, this is necessary to see the fluorescent signals from virions and antibodies (Figs 1,2,3 and 5). However, this makes it difficult to see the virions in the sections. The small white arrows are likely pointing to virions (I presumed this as I did not see where it was stated in text or figure legends) yet seeing them by eye still proved difficult. As the graphs in figures 4 and 5 are dependent on these data this difficulty reduces confidence in the graphs and therefore in the overall conclusions from the manuscript.

2. The background green fluorescence seems high. Can the background fluorescence be turned down so that we can better see the IgA or the virions?

3. A number of interesting and unexpected attributes were associated with the IgA administration. Is there any way to quantify the frequency of virions that are near IgA staining and to determine if IgA binding is influencing the depth of penetration or proportion penetrating.

4. Is there any way to demarcate the edges of the epithelium in the rectal images so that we can more easily see the depth by eye?

5. Table does not make clear which of the macaques are shown in figure 2 (with the exception of A15X085, which is stated in Table 1 as being evaluated by CT but discussed in text as being evaluated by PET??)

Another major issue is whether there are any important differences associated with utilizing either PET or CT? These are discussed interchangeably in the text, but are the results actually interchangeable, or were there some interesting differences that could be discussed? Is one method superior to the other?

The finding of the virus at the mesenteric lymph nodes at 2 hours post viral administration is particularly interesting. Is there any way to evaluate which of the lymph node areas (T or B cells) is the location where the virions are being observed? Could you do immunofluorescence analysis with anti-CD3, CD4 or CD20 on adjacent section of the tissue? Was there any evidence for IgA being seen associated with those virions in the lymph node?

**Part III – Minor Issues: Editorial and Data Presentation Modifications**

Reviewer #1: Please explain in all figure legends what arrows are pointing to.

The authors refer to rectal challenge studied in mice for the murine leukemia virus (line 401), but I couldn’t find it in the cited paper. The paper describes s.c., i.v., and oral transmission, but not rectal challenge. It does describe virus arrival at mesenteric lymph nodes after oral uptake.

Reviewer #2: Line 62 should read "Currently, an effective human immunodeficiency virus (HIV) vaccine does not exist".

Line 104: "Notably, most vaccines do not stimulate protective IgA responses". Do we really know this, or do we assume this because IgA is seldom studied in vaccine trials?

Line 229 - a word seems to be missing ("to consider whether the entire colon needs to be excised")

Line 262 - for clarity, text should read "but Cy5 signal was apparent in rectum, descending..."

Lines 372-375, wording could be clearer.

Line 381: predominantly

Lines 407-9: Should read "while HGN194 dIgA2 protected 1/6 animals"

Lines 413-414: "The issue...would be moot (not mute) if the virions within the immune complexes are inactivated or neutralized".

Reviewer #3: Some of the minor issues:

1. What is meant by this sentence at line 386: “It may be possible that dIgA2 has a greater capacity to form immune complexes with pathogens in the colon, specifically for shuttling to mesenteric lymph nodes and further inside the colon than dIgA1.”? Are you saying that immune complexes with IgA1 are not forming due to the environment (pH manybe)? Could it be that immune complexes are forming but receptors for the antibody are not present?

2. Figure 4, text in manuscript is listed as A-C, Figure has A-I.

3. Line 408, HGN194dIgA1 is repeated, I think one should be A2

4. Line 423, Possibly include a discussion of the finding from the STEP HIV vaccine trial where they observed a correlation between serum IgA levels and increased risk of HIV transmission in those that were vaccinated.

5. Line 735, typo ‘dD’

PLOS authors have the option to publish the peer review history of their article (what does this mean?). If published, this will include your full peer review and any attached files.

Reviewer #1: No

Reviewer #2: No

Reviewer #3: No
---

## [Decision Letter · Decision Letter 1]

11 May 2021

Dear Professor Hope,

We are pleased to inform you that your manuscript 'PET/CT targeted tissue sampling reveals virus specific dIgA can alter the distribution and localization of HIV after rectal exposure.' has been provisionally accepted for publication in PLOS Pathogens.

Before your manuscript can be formally accepted you will need to complete some formatting changes, which you will receive in a follow up email. A member of our team will be in touch with a set of requests. Also, please adjust the arrows on the figures as requested by Reviewer #1: "placement of the arrows in all figures still remains an issue. They should be placed consistently exactly 1 mm from the event, not shifted left or right."

Best regards,

Daniel C. Douek

Associate Editor

PLOS Pathogens

Richard Koup

Section Editor

PLOS Pathogens

Kasturi Haldar

Editor-in-Chief

PLOS Pathogens

orcid.org/0000-0001-5065-158X

Michael Malim

Editor-in-Chief

PLOS Pathogens

orcid.org/0000-0002-7699-2064

Reviewer Comments (if any, and for reference):

Reviewer's Responses to Questions

**Part I - Summary**

Reviewer #1: The authors have addressed most of my concerns, but the placement of the arrows in all figures still remains an issue. They should be placed consistently exactly 1 mm from the event, not shifted left or right. This remains a problem.

Reviewer #3: Changes to manuscript have addressed my previous issues. Microscopic images have been improved and virions are now more easy to visualize. Manuscript addresses an important question regarding role for antibody in HIV infection and spread.

**Part II – Major Issues: Key Experiments Required for Acceptance**

Reviewer #1: None

Reviewer #3: (No Response)

**Part III – Minor Issues: Editorial and Data Presentation Modifications**

Reviewer #1: (No Response)

Reviewer #3: (No Response)

PLOS authors have the option to publish the peer review history of their article (what does this mean?). If published, this will include your full peer review and any attached files.

Reviewer #1: No

Reviewer #3: No

---

## [Editor Report · Acceptance letter]

22 May 2021

Dear Professor Hope,

We are delighted to inform you that your manuscript, "PET/CT targeted tissue sampling reveals virus specific dIgA can alter the distribution and localization of HIV after rectal exposure.," has been formally accepted for publication in PLOS Pathogens.

Best regards,

Kasturi Haldar

Editor-in-Chief

PLOS Pathogens

orcid.org/0000-0001-5065-158X

Michael Malim

Editor-in-Chief

PLOS Pathogens

orcid.org/0000-0002-7699-2064